# Does Adult Children’s Education Increase Parents’ Longevity in China?

**DOI:** 10.3390/ijerph192315530

**Published:** 2022-11-23

**Authors:** Yanna Ma, Zhanli Ma, Moqin Yang

**Affiliations:** 1School of Economics, Central University of Finance and Economics, Beijing 100081, China; 2School of Statistics and Mathematics, Zhongnan University of Economics and Law, Wuhan 430073, China

**Keywords:** adult children’s education, parents’ longevity, truncated regression, emotional support

## Abstract

The populations of some developing countries are aging rapidly, while the average years of schooling for residents are also constantly increasing. However, the question of whether adult children’s education affects the longevity of their older parents remains understudied. This study used China Health and Retirement Longitudinal Survey data to estimate the causal impact of adult children’s education on their parents’ longevity using a truncated regression model. We found that, for every one-unit increase in adult children’s education, fathers’ and mothers’ longevity increases by 0.89 and 0.75 years, respectively. The mechanism analysis shows that adult children’s education significantly increases their provision of emotional and financial support to their parents, as well as their parents’ self-reported health. Further evidence shows that for every one-unit increase in adult children’s education, the longevity of their fathers-in-law and mothers-in-law also increases by 0.40 and 0.46 years, respectively. Improving the level of adult children’s education can thus increase parents’ and parents-in-law’s longevity via three channels: providing emotional and financial support and improving health. A culture in which parents value their children’s education should thus be promoted.

## 1. Introduction

The intergenerational transmission of human capital between parents and children has long been regarded as a “one-way street.” Economists and sociologists have mainly considered the impact of parents’ human capital and socioeconomic status on their children [1]. However, the question of whether the human capital of adult children “pays back” their parents as they age deserves further study from two perspectives.

First, the Chinese population is aging rapidly. It is predicted that the population over 65 years old in China will peak between 2030 and 2050 [2]. This aging population is exacerbating the problem of supporting the elderly. Based on social exchange theory [3], parents allocate resources to their children in their early years and adult children give back some of their resources to parents in later lifecycle stages. Since children’s education is an important way to strengthen the concept of filial piety and increase income, as filial piety and children’s earnings directly affect the quality of their parents’ care [4], the impact of adult children’s formal educational attainment on their parents’ health and longevity cannot be ignored.

Second, with the expansion and popularization of the compulsory education system in China, years of schooling for Chinese people are constantly increasing. In 2020, the average years of education of China’s population aged 15 years and above reached 10.75 years—an increase of 1.08 years over 2010. The illiteracy rate also dropped from 4.08% in 2010 to 1.67% in 2020. The fact that society and family attach great importance to children’s education has meant that children now receive significantly more schooling than their parents did. According to data from the China Health and Retirement Longitudinal Survey (CHARLS) [5], the average number of years of education of parents is 1.31 compared with 5.33 for their children. In this context, analyzing the impact of adult children’s education on the longevity of their parents is of great significance for improving the quality of life of the elderly.

Education in this study means formal education, a structured and systematic form of learning; generally, this is education of a certain standard delivered to students by trained teachers in a designated place of education. Education is an important factor influencing an individual’s longevity, affecting their health and chances of living a healthy life [6,7]. Some recent studies have shed light on whether children’s education affects parents’ longevity, especially as parents age and need more care. Most examine the effect of children’s education on the probability of their parents dying or surviving. De Neve and Fink [8] used a comparative analysis before and after an education reform in Tanzania and found that the improvement in children’s education level can significantly increase the probability of parents over 50 years old surviving. Torssander [9] compared the different longevity of siblings using Swedish data. The results showed that sisters with children who have only received compulsory education are likely to die younger than sisters with children who have received higher education. Moreover, adult children’s education affects their parents’ probability of both falling ill and dying after illness. Zimmer et al. [7], using data from Taiwan, showed that children’s education has a significant effect on reducing parents’ mortality, especially for elderly parents with serious illnesses. Friedman and Mare [10], using data from the United States, found that highly educated children increase the longevity of their parents, especially by reducing the risk of them dying from chronic respiratory and lung diseases. De Neve and Harling [11] found that children’s increasing educational attainment is associated with a decline in the risk of parental death in South Africa.

However, some studies have drawn different conclusions. Lundborg and Majlesi [12], using the instrumental variable approach, argued that, while the implementation of compulsory education has increased years of schooling, this policy has not significantly increased the life expectancy of parents, as children’s education level increases the distance between parents’ and children’s residences. This increased distance makes it less likely that children will live with their parents, which reduces parents’ investment in their own health. Therefore, previous studies have drawn inconsistent conclusions about the effect of adult children’s education on parents’ longevity.

Building on previous work, the first hypothesis is as follows:

**Hypothesis** **1.***Adult children with higher education increase the longevity of their parents*.

Previous studies that have explored the impact of children’s education on parents’ longevity suffer from two main shortcomings. First, an analysis of the possible influencing mechanisms of this impact is lacking. Second, the possible problem of endogeneity has not been adequately addressed. To address the above shortcomings, this study employed CHARLS data to empirically analyze the effect of children’s education on their parents’ longevity in China. We also explored the possible mechanisms from the perspective of the theoretical and empirical evidence. We show that adult children’s education extends parents’ longevity significantly, a finding that remains valid after conducting robustness tests and running an instrumental variable regression. In particular, we find that enhancing the level of children’s education can increase the emotional and financial support they provide to their parents, as well as improve their parents’ health. These three channels are critical to prolonging the longevity of elderly parents. The heterogeneity analysis indicates that the effect of children’s education on parents’ longevity is more obvious in rural areas and in families with multiple children. Further analysis shows that increasing the education level of individuals also extends the longevity of parents-in-law.

Compared with the existing literature, this study makes three novel contributions. First, examining the influence of adult children’s education on parents’ longevity not only helps understand the influencing factors of parents’ longevity, but also provides a new solution to the problem of supporting the elderly. Second, summarizing and refining three mechanisms through which adult children’s education affects parents’ longevity using detailed microdata on China provides a solid basis for solving the global problem of the extent to which it is appropriate for families to support the elderly. Finally, this study analyzes the influence of adult children’s education on parents-in-law’s longevity. This provides strong support for the need to reconsider family relations and promote family harmony in the context of the aging population and imperfect social security system in some developing countries such as China.

The rest of this paper is organized as follows. Section 2 outlines the possible channels linking the longevity of older parents with adult children’s education Section 3 introduces detailed information about the data, variables, and model used. Section 4 reports the empirical results for parents’ longevity and the underlying mechanisms. It also checks the robustness of the results. Section 5 discusses the findings of the study and concludes.

## 2. Theoretical Analysis

Social network theory [13] suggests that social networks impact health through four primary mechanisms: the provision of social support, social influence, social engagement and attachment, and access to resources and material goods. Hence, social network theory and social exchange theory [3] suggest that three mechanisms drive the influence of adult children’s education on their parents’ longevity: providing emotional and financial support and enhancing parents’ health.

First, recent studies show that children with more years of formal schooling provide more emotional support to their older parents, which improves parents’ mental health. For example, such children are more likely to increase their frequency of contact and strengthen communication with their parents [14]. Spitze and Logan [15] found that, at the same level of educational attainment, adult daughters contact their parents more frequently than do adult sons. They provide emotional support by protecting their parents from stressful events, supporting their good mental health, and strengthening their social status [16]. Further, adult children are more likely to enjoy stable working and living environments when they are highly educated, which reduces parents’ worries about their children and mitigates the negative impact on parents’ mental health [17]. However, some studies have highlighted that highly educated children incur a higher opportunity cost of contact with their parents, meaning that they may provide less emotional support [18]. In summary, as previous researchers have arrived at different conclusions on the impact of adult children’s education on providing their parents with emotional support, further research is required on the workings of this mechanism in the context of traditional Chinese culture.

In light of the discussion above, the second hypothesis was developed as follows:

**Hypothesis** **2.***The association between having highly educated adult children and parents’ longevity is partly explained by the emotional support provided by adult children*.

Second, based on social exchange theory [3], an improvement in children’s education can increase the financial support they provide to their parents. Adult children who have higher education can be more willing and able to support their elderly parents, thereby improving the quality of care for older parents through current and long-term intergenerational support [4]. Moreover, to compensate for their parents’ investment in their education when they were young, adult children with higher education tend to provide more financial support to their elderly parents [19]. In addition, in rural areas, the relationship between parents and adult children’s support is characterized by reciprocity. There is a significant positive correlation between parents’ investment in their children’s education and the old-age return provided by adult children [20]. Investment in children’s education in the early stage of children’s lives can improve the quality of life of parents once they pass the age of 65. The longer children’s education period, the better the quality of life of parents as they age [21]. Moreover, highly educated adult children are motivated to help other family members and tend to share their resources with their parents. Indeed, even if adult children do not live with their parents, those who have a higher income and/or socioeconomic status due to a higher education provide more financial support to their parents [9]. Therefore, the higher the level of education of adult children, the more likely they are to provide more financial support.

In light of the discussion above, the third hypothesis was developed as follows:

**Hypothesis** **3.***The association between having highly educated adult children and parents’ longevity is partly explained by the financial support provided by adult children*.

Finally, the educational attainment of adult children is conducive to improving parents’ health. On the one hand, the increase in children’s human capital can help parents process health and disease information more correctly [8]. In other words, the improvement in cognitive ability can be shared with parents. Adult children who acquire health knowledge or display healthier lifestyles can influence parents’ attitudes toward healthy behaviors and lifestyles, including improving the residential environment and reducing smoking and alcohol abuse [22]. This also increases the possibility of parents participating in sports and physical activities [23]. On the other hand, higher education of adult children directly improves the cognitive ability of elderly parents, prevents the occurrence of Alzheimer’s disease, improves the function of the instrumental activities of daily living, reduces the incidence of chronic diseases, and enhances social adaptability [16,24]. When parents are sick, adult children with higher education can provide better medical resources and care, thereby increasing the probability of their parents’ recovering and improving their health in the long run [7]. Therefore, adult children with higher education can help their parents acquire good health habits, display more positive health behaviors, and reduce the probability of illness.

In light of the discussion above, the fourth hypothesis was developed as follows:

**Hypothesis** **4.***The association between having highly educated adult children and parents’ longevity is partly explained by parents’ health*.

## 3. Data and Methodology

### 3.1. Data Source and Variables

The main data source of this study was the CHARLS conducted in 2011, 2013, 2015, and 2018, with regional data drawn from the Chinese Research Data Services Platform and China Urban Statistical Yearbook. The CHARLS investigates elderly families and their members aged 45 and above. Hence, all the adult children surveyed (i.e., our respondents) are aged over 45 years. It provides abundant information on individuals and their families, including parents and children. In addition, it describes in detail the age at which parents died, time of death, number of children, and intergenerational support between children and parents. Of this information, this study extracts the age at which parents died to calculate their longevity. These data are finally integrated into cross-sectional data, which include information on parents’ and children’s characteristics. This dataset includes 10,525 observations of fathers and 11,344 observations of mothers.

The dependent variables in this study are mothers’ longevity and fathers’ longevity (i.e., the age at which they died). If the father or mother was still alive in 2018, their age in that year was used. Because the survey respondents are all over 45 years old in 2011, their parents who were still alive in 2018 (the most recently conducted survey wave) were 72 years old and above. This suggests that these parents have lived a long life.

The core independent variable is adult children’s education, which is divided into five levels with values of 1–5, respectively: illiteracy, primary school, junior high school, high school (and vocational education), and junior college and above. Years of education are used in a robustness test. Referring to the assignment of education level by Yang et al. [25], illiteracy, primary school, junior high school, high school (and technical secondary school), junior college, university, postgraduate, and doctoral level are designated as 0, 6, 9, 12, 15, 16, 19, and 22 years of schooling, respectively.

The control variables include fathers’ and mothers’ occupation prestige, whether parents have sons or daughters, the number of children, whether respondents own real estate, property value, and whether they live in urban areas. Occupation prestige refers to the occupation prestige ranking proposed by Li [26]. This study orders the occupations in the survey from low to high and divides them into seven types. From low to high, these are unemployment (unclassified occupation or no occupation), farmers (production worker in the agriculture, forestry, animal husbandry, fishery, and water conservation industries), workers (production and transportation equipment operators and related workers), business and service workers, clerks (clerks and related workers), professional and technical workers, and managers (heads of enterprises, including state-owned enterprises, and institutions). These seven types of occupations are assigned values of 1–7, respectively.

The number of children was assigned values from 1–10; if the number of children is more than 10, it was coded 10. If the parents have sons or daughters, the presence of a child was coded 1, and 0 otherwise. If the surveyed adult children live in urban areas, this indicator was coded 1; it was coded 0 for rural areas. If the surveyed children have real estate, this indicator was coded 1, and 0 otherwise. The property value was taken as the logarithm of the property value in 2011. Finally, to control for any birth cohort effect, parents’ birth cohort was divided into six groups: before 1900, 1901–1910, 1911–1920, 1921–1930, 1931–1940, and 1941–1950.

The provision of emotional support and financial support to parents, as well as parents’ health, were measured as follows. Emotional support was measured by the frequency of visiting parents, divided into almost never, once a year, once every six months, once every three months, once every month, once every fortnight, once every week, two to three times a week, and almost every day. The values were 1–9, respectively. The higher the value, the higher the frequency of visiting parents. If adult children live with their parents, the value was 9. The provision of financial support to parents was measured by the sum of the value of money and items given to their parents regularly or irregularly. To reduce heteroscedasticity, the logarithm of financial support was used. Parents’ health was self-reported on a scale from 1 to 5. The higher the value, the better their health. Table 1 shows the descriptive statistics.

### 3.2. Empirical Model

Our empirical analysis used a truncated regression model. This model uses nonparametric analysis to mitigate the problem of estimation bias caused by parametric regression. Hence, this model is usually suitable for dependent variables with limited censoring. As noted above, the dependent variables were fathers’ longevity and mothers’ longevity. Parents can obtain emotional and financial support from their children if and only if they are at least 40 years of age and above when they die. In addition, despite scientific and technological developments, human lifespan cannot be extended indefinitely. The range of dependent variables is thus limited. The adult children’s ID served as the unit of analysis. Different adult children’s parents have different IDs, even if they have the same parents. The truncated regression model was set as follows:(1)$$longevity_{i}^{*}=\alpha_{0}+\alpha_{1}edu_{i}+\delta X_{i}+cohort_{i}+community_{i}+\varepsilon_{i}$$
$$longevity_{i}=\left\{\begin{array}{c} longevity_{i}^{*},\;\;\mathrm{if}\;longevity_{i}^{*}\geq 40\\ 40\;\;\;\;\;\;\;\;\;\;\;\;\;\;\;\;,\;\;\mathrm{if}\;longevity_{i}^{*}<40 \end{array}\right..$$
where $longevity_{i}^{*}$ is the longevity of the father or mother of individual *i*. When longevity is greater than or equal to 40 years, this variable can be observed; however, when longevity is less than 40 years, there is no information on this observation. The variable $edu_{i}$ is the education level of individual *i*. $\alpha_{1}$ is the parameter of interest, which measures the effect of increasing adult children’s years of schooling by one year on their parents’ longevity. $cohort_{i}$ is the birth cohort fixed effect. $community_{i}$ is the community fixed effect. $\varepsilon_{i}$ is the random disturbance term. Standard errors are clustered at the community level. $\;X_{i}$ represents the control variables such as parents’ and children’s characteristics.

Furthermore, to verify the goodness-of-fit of the above model, the Cox proportional hazards model was used in a robustness check. This model adopts semi-parametric estimation and does not make fixed restrictive assumptions about the benchmark risk function. In our study, this model was used to estimate the hazard ratios of parents’ longevity when their adult children have different levels of education. Parental age in years is the time variable and the death of a parent is an event. Censoring occurred when the parents died, or in 2018. Parents entered our analysis if they were over 40 years old when they died. We again used community fixed effects. Standard errors were clustered at the community level. The model was set as follows:(2)$$\;\;h\left(t\right)=h_{0}\left(t\right)e^{\left(\beta X\right)}.$$
where $h\left(t\right)$ indicates the probability of death at t years old under the condition that the individual has survived until *t*−1 years old. $\;h_{0}\left(t\right)$ is the parent’s benchmark mortality risk function when there is no covariate and X is the variable affecting parents’ longevity (i.e., all the independent variables in Equation (1)).  β represents the vectors of the parameters to be estimated. The dependent variable is a dummy variable of whether the father or mother died. If the father or mother survived until age *t*−1 and died at the age of *t*, the father or mother is considered to be dead.

## 4. Results

### 4.1. Fathers’ Longevity

Table 2 shows the results of the impact of adult children’s education on fathers’ longevity, estimated using the truncated regression model. Column 1 in Table 2 shows that adult children’s education significantly increases fathers’ longevity. After adding fathers’ education and occupation prestige, the results in column 3 show that fathers’ education significantly raises their longevity. The improvement in one’s education not only increases one’s own resources, but also improves the productivity of one’s health investment, which increases longevity. Similarly, fathers’ occupation prestige significantly increases longevity by improving income and social capital. Column 5 shows that, when children own real estate, this significantly increases their fathers’ longevity. The results of the marginal effect estimation in column 6 show that fathers’ longevity increases by 0.89 years on average when their adult children’s education increases by one year of schooling.

### 4.2. Mothers’ Longevity

Table 3 shows the estimation results of mothers’ longevity using the same truncated regression model. After adding the birth cohort and community fixed effects, the results in column 1 show that adult children’s education significantly increases mothers’ longevity. The control variables of parents’ and children’s characteristics are added into columns 3 and 5, respectively. The results continue to show that adult children’s education significantly increases mothers’ longevity. In particular, column 5 shows that mothers’ education and occupation prestige both significantly increase their longevity. Similarly, an increase in the number of children and the presence of both sons and daughters significantly increase mothers’ longevity. The marginal effect analysis in column 6 shows that, on average, a one-unit increase in adult children’s education extends mothers’ longevity by 0.75 years. Table 2 and Table 3 therefore show that adult children’s education has a positive spillover effect on parents’ longevity.

### 4.3. Instrumental Variable Analysis

The above estimation may suffer from endogeneity problems. First, there may be missing variables. Other factors affecting longevity, in addition to those factors included in the above regression, could also be present. These include factors such as genes, parents’ eating habits, and family harmony, which cannot be controlled for in this study. The reverse causality problem may also exist. The longer the life expectancy of parents, the more likely they are to invest in their children’s education and improve their children’s human capital to receive more support from their adult children as they age.

To address these possible endogeneity problems, this study used an instrumental variable regression. Following the instrumental variable estimation used by Chen et al. [27], the per capita number of Jinshi (Jinshi was a successful scholar in the Qing Dynasty of Chinese history) in the Qing Dynasty was used as an instrumental variable of adult children’s education. Historically, the greater the per capita number of Jinshi in cities, the more local citizens invest in children’s education and, thus, the greater their children’s access to education. However, the per capita number of Jinshi in cities does not directly impact longevity.

Due to the availability of data, the population of cities in 1984 was used to proxy for their population in the Qing Dynasty. The per capita number of Jinshi was obtained by dividing the number of Jinshi by the population in this region. The two-stage least squares method was used to estimate the instrumental variables. The results of the Hausman test showed that adult children’s education was rejected as an exogenous variable at the 1% level. The one-stage regression results in columns 1 and 3 of Table 4 show that the per capita number of Jinshi in the Qing Dynasty significantly increases adult children’s education. The higher the per capita number of Jinshi in the Qing Dynasty, the higher the adult children’s education. Columns 2 and 4 in Table 4 show that, in the two-stage least squares regression with fathers’ longevity and mothers’ longevity as the dependent variables and the control variables of parents’ and children’s characteristics, adult children’s education significantly increases parents’ longevity. Hence, the regression results when dealing with endogeneity still support the main finding that adult children’s education significantly increases parents’ longevity.

### 4.4. Robustness Check

To test the robustness of the above results, we used three alternative estimation methods: bootstrapping, the Cox proportional hazards model, and multiple linear regression. First, the bootstrapping method was used to select the sample 500 times randomly from all the observations and, then, we used the truncated regression model to fit parents’ longevity. Columns 1 and 7 in Table 5 show that, after using the bootstrapping method, the results still support the fact that adult children’s education significantly improves parents’ longevity. Columns 2 and 8 were estimated using multiple linear regression, and the results still support the above outcomes. Finally, we used the Cox proportional hazards model to estimate the influence of adult children’s education on parents’ mortality risk. The results in columns 3 and 9 show that a one-year increase in adult children’s education reduces fathers’ mortality risk by 0.058 times and mothers’ mortality risk by 0.062 times. This result shows that an increase in adult children’s education can significantly reduce parents’ mortality risk.

To estimate the impact of adult children’s education on parent’s longevity more accurately, we excluded parents who are alive in 2018 (i.e., the data only contain deceased parents). The results in columns 4 and 10 of Table 5 show that, after controlling for parents’ and children’s characteristics and including the birth cohort and community fixed effects, adult children’s education still significantly increases parents’ longevity. The marginal effect analysis shows that, on average, a one-year increase in adult children’s education increases fathers’ longevity by 0.65 years and mothers’ longevity by 0.64 years. In columns 5 and 11, the degree of adult children education is converted into years of education under the current education system. The marginal effect analysis shows that, on average, increasing adult children’s education by one year raises fathers’ longevity by 0.27 years and mothers’ longevity by 0.20 years. In columns 6 and 12, the degree of adult children’s education is replaced by the highest degree of education among parents’ children. The result still supports our main conclusions. Overall, the results of the baseline truncated regression model thus remain robust after using multiple linear regression, the bootstrapping method, and the Cox proportional hazards model. That is, the results still support the conclusion that adult children’s education significantly increases parents’ longevity.

### 4.5. Mechanism Analysis

According to the theoretical analysis above, providing emotional and financial support to parents and parents’ health are the three main mechanisms of the impact of children’s education on parents’ longevity. Table 6 shows the estimation results of our examination of these mediating mechanisms. Columns 1 and 4 show that adult children’s education significantly increases the provision of emotional support to parents. This indicates that the higher adult children’s education, the higher the frequency of contacting parents. Columns 2 and 5 show that adult children’s education significantly increases the provision of financial support to parents. This demonstrates that the higher adult children’s education, the more willing they are to provide financial or material help to their parents. Column 6 shows that the increase in adult children’s education significantly improves mothers’ self-reported health. In conclusion, the provision of emotional and financial support and parents’ health all serve as channels. Adult children’s education influences parents’ longevity by allowing them to provide their parents with more emotional and material help as well as improving their parents’ health.

### 4.6. Heterogeneity Analysis

As individuals, families, and regions differ markedly, our study analyzed the heterogeneity of different groups. Our sample was divided based on whether respondents live in urban areas and whether the number of children is more than two. First, accessibility to and the abundance of medical resources differ between urban and rural areas. Second, adult children can provide different resources to their parents. On the one hand, the more children older parents have, the more support they have. On the other hand, in families with many children, when supporting older parents, it is likely that the children will shift the responsibility onto each other. The resultant effects should also be verified based on the background of Chinese culture.

All the control variables were added into these regressions, along with the birth cohort and community fixed effects. First, column 1 in Table 7 shows that the cross-multiplication term between urban areas and adult children’s education is significantly less than 0, which indicates that adult children’s education in rural areas significantly increases fathers’ longevity more than that in urban areas. For children with higher education in rural areas, the health spillover effect is higher. There are many left-behind elderly people in rural areas of China. Hence, increasing adult children’s education is also beneficial to prolonging the longevity of parents.

Second, columns 2 and 4 show that the cross-multiplication term between the number of children and their education is significantly positive. This indicates that when there are more than two adult children, the probability of them contacting their parents rises. Therefore, the financial and emotional support parents receive from their adult children rises, thereby increasing older parents’ longevity. Thus, the positive influence of adult children’s education on parents’ longevity is greater for those who have more children and whose adult children live in rural areas.

### 4.7. Parents-in-Law’s Longevity

Guan [28] found that daughters-in-law and sons-in-law feel differently toward their parents-in-law than toward their biological parents. We thus analyzed the influence of adult children’s education on the longevity of their in-laws. The results in columns 1 and 2 of Table 8 show that, on average, a rise in adult children’s education by one year of schooling increases the longevity of fathers-in-law by 0.37 years. Similarly, columns 3 and 4 show that, on average, a rise in adult children’s education by one year of schooling increases the longevity of mothers-in-law by 0.4 years. Hence, the improvement in adult children’s education has a positive spillover effect not only on their parents’ longevity, but also on their spouse’s parents’ longevity.

## 5. Discussion and Conclusions

Using CHARLS data, this study examined the effect of adult children’s education on parents’ longevity using the truncated regression model in addition to the Cox proportional hazards model, instrumental variable estimation, tests of the mediating effect, and related robustness tests. We found that improving the level of adult children’s education has a positive spillover effect on parents’ longevity. Specifically, increasing adult children’s education by one unit significantly increases fathers’ longevity by 0.89 years and mothers’ longevity by 0.75 years. In addition, we find that the spillover effect of adult children’s education on parents’ longevity manifests in three ways. Adult children with higher education increase the emotional and financial support they provide to their parents, as well as improving their parents’ health. In particular, the increased frequency with which well-educated adult children contact their parents leads to improvements in their parents’ mental and physical health; such children can also use their resources to provide high-quality medical services to their older parents. This behavior ultimately prolongs their parents’ longevity. The heterogeneity analysis found that the positive influence of adult children’s education on parents’ longevity is greater in rural areas and for families with more children. Further analysis found that improving the level of adult children’s education also had a positive spillover effect on parents-in-law’s longevity.

Many studies have explored the relationship between education and health [6,29]. At the same time, several economics studies have suggested that children’s education has a positive spillover effect on parents’ health [20,30]. This effect exists not only in China, but also in the United States [10], South Africa [11], and Europe [20]. Our research adds to the literature on the relationship between adult children’s human capital and parents’ health, and the results show that the intergenerational transfer between children and parents does not flow unidirectionally. Improving the degree of children’s education not only benefits children, but also benefits their parents.

The results of our study extend previous research on the relationship between children’s education and parents’ health or mortality risk [20,30]. One previous study examined the effect of children’s higher education on the mortality rate of parents with a low education level, concluding that the positive spillover effect of children’s education on younger elderly people was greater than that on older elderly people [20]. However, our research not only demonstrates the spillover effect of adult children’s education onto parents’ longevity but also concludes that the improvement in adult children’s education is beneficial to extending the longevity of parents-in-law. This finding encourages parents-in-law, who are an important part of the family, to connect with their sons- and daughters-in-law more often.

A limitation of our research is that the quality of the education of adult children was not considered because of the availability of data. The quality of adult children’s education is another important factor affecting parents’ longevity. The quality of the school attended by the adult children affects the quality of education. However, the name of the school attended is unavailable in our data. The second limitation is the lack of analysis of regional differences in living standards and life expectancy. In our data, parents’ birth dates range between 1900 and 1950, a period in which some regions experienced war. However, the effect of the experience of war on parents’ lifespan was not considered in our study. Meanwhile, De Neve and Fink [8] stated that the spillover effect of education on health differs by region. The third limitation is that we did not examine which of the three mechanisms is the most important; moreover, no other possible mechanisms were explored.

Nonetheless, the conclusions of this study suggest important policy implications. First, adult children’s education is an important factor affecting their parents’ longevity. A culture in which parents value their children’s education should thus be promoted. When a culture of higher education for children is cultivated, children are more likely to be willing to take on the responsibility of supporting their parents as they age. This can not only reduce endowment insurance pressure in China, but also improve the quality of parents’ lives as they become old. Second, family harmony should be promoted and a good family atmosphere should be created. Adult children’s education is beneficial to their spouse’s parents’ longevity, in addition to their own parents’ longevity. Parents are more likely to follow their children’s advice to increase their own healthy behaviors, instead of following the advice of other relatives. Therefore, family harmony should be advocated to increase the emotional connection between children and their parents. Finally, the elderly should be given training and support on the use of the mobile Internet. In the context of global aging, to reduce the gap between life expectancy in rural and urban areas and improve the quality of life among the left-behind elderly, access to and the usage of the Internet are important. Increasing network use training for the elderly would ensure that migrant children and the left-behind elderly can communicate via video when they are unable to meet frequently. As noted earlier, children increasing the emotional support they provide to their parents helps improve the latter’s physical and mental health. Education thus represents the distribution of family resources and is a means of addressing inequalities in family health. The increase in the education level of adult children can mitigate their parents’ lack of financial resources.

Further research should pay more attention to the extent to which the socioeconomic status of adult children affects their parents’ longevity and investigate how such an effect might differ across regions and populations, as well as over time.

## Figures and Tables

**Table 1 ijerph-19-15530-t001:** Descriptive statistics of the variables.

Variable	Definition	Observations	Mean	Standard Deviation	Min	Max
Fathers’ longevity	Fathers’ lifespan	10,525	71.904	13.557	40	107
Mothers’ longevity	Mothers’ lifespan	11,344	75	13.354	40	106
Fathers’ education	Fathers’ education level	10,525	0.472	0.745	0	5
Mothers’ education	Mothers’ education level	11,344	0.467	0.751	0	5
Fathers’ year of birth	Year of fathers’ birth	10,525	1921.608	13.039	1830	1950
Mothers’ year of birth	Year of mothers’ birth	11,344	1923.993	12.654	1830	1950
Children’s education	Education level of respondents	11,344	1.213	1	0	5
Fathers’ occupation prestige	Fathers’ occupation prestige ranking	11,344	2.793	1.669	1	7
Mothers’ occupation prestige	Mothers’ occupation prestige ranking	11,344	2.101	0.803	1	7
Real estate	Whether respondents own real estate	11,344	0.902	0.297	0	1
Property value	Natural logarithm of the value of respondents’ property	11,344	1.932	1.574	0	13.592
Number of children	Number of children of parents	11,344	3.604	2.398	1	14
Whether parents have sons	At least one son among the children of parents	11,344	0.935	0.246	0	1
Whether parents have daughters	At least one daughter among the children of parents	11,344	0.932	0.252	0	1
Urban area	Whether respondents live in urban areas	11,344	0.398	0.489	0	1
Emotional support	Frequency of visiting parents	3220	6.069	2.262	1	9
Financial support	Natural logarithm of transfer payments to parents	3220	3.205	2.971	0	10.309
Mothers’ health	Mothers’ self-reported health in 2011	2708	2.811	0.963	1	5
Fathers’ health	Fathers’ self-reported health in 2011	2243	2.959	0.985	1	5

Note: Every adult child has a unique ID number.

**Table 2 ijerph-19-15530-t002:** Estimation results of adult children’s education on fathers’ longevity.

Variable	Coefficient	Marginal Effect	Coefficient	Marginal Effect	Coefficient	Marginal Effect
	(1)	(2)	(3)	(4)	(5)	(6)
Children’s education	1.081 ***	1.017 ***	0.975 ***	0.917 ***	0.943 ***	0.887 ***
	(0.152)	(0.142)	(0.152)	(0.142)	(0.152)	(0.143)
Fathers’ education			0.888 ***	0.835 ***	0.887 ***	0.834 ***
			(0.214)	(0.201)	(0.214)	(0.201)
Fathers’ occupation prestige			0.276 ***	0.259 ***	0.266 ***	0.250 ***
			(0.089)	(0.084)	(0.089)	(0.084)
Fathers’ year of birth			−0.158 ***	−0.148 ***	−0.161 ***	−0.151 ***
			(0.040)	(0.037)	(0.040)	(0.037)
Number of children					0.093	0.087
					(0.063)	(0.059)
Whether parents have sons					0.277	0.260
					(0.614)	(0.577)
Whether parents have daughters					0.702	0.661
					(0.544)	(0.512)
Urban area					1.473 ***	1.386 ***
					(0.362)	(0.341)
Real estate					0.973 *	0.915 *
					(0.529)	(0.498)
Property value					0.160	0.150
					(0.116)	(0.109)
Constant	70.88 ***		369.67 ***		373.05 ***	
	(0.560)		(75.186)		(74.900)	
Birth cohort fixed effect	Yes	Yes	Yes	Yes	Yes	Yes
Community fixed effect	Yes	Yes	Yes	Yes	Yes	Yes
Observations	10,525	10,525	10,525	10,525	10,525	10,525

Note: The data in brackets are standard deviation, and *, and *** represent significance at the 10%, and 1% levels, respectively. Columns 1, 3, 5, and 7 show the estimation coefficients of the truncated regression model, and columns 2, 4, 6, and 8 show the average marginal effect.

**Table 3 ijerph-19-15530-t003:** Estimation results of adult children’s education on mothers’ longevity.

Variable	Coefficient	Marginal Effect	Coefficient	Marginal Effect	Coefficient	Marginal Effect
	(1)	(2)	(3)	(4)	(5)	(6)
Children’s education	0.822 ***	0.797 ***	0.828 ***	0.803 ***	0.774 ***	0.751 ***
	(0.143)	(0.138)	(0.142)	(0.138)	(0.143)	(0.139)
Mothers’ education			0.470 *	0.456 *	0.488 *	0.474 *
			(0.257)	(0.250)	(0.257)	(0.249)
Mothers’ occupation prestige			0.317 *	0.307 *	0.325 **	0.315 **
			(0.162)	(0.157)	(0.163)	(0.158)
Mothers’ year of birth			−0.152 ***	−0.147 ***	−0.158 ***	−0.153 ***
			(0.040)	(0.039)	(0.040)	(0.039)
Number of children					0.198 ***	0.192 ***
					(0.055)	(0.054)
Whether parents have sons					1.726 ***	1.674 ***
					(0.579)	(0.562)
Whether parents have daughters					1.497 ***	1.453 ***
					(0.554)	(0.537)
Urban area					3.919 ***	3.803 ***
					(0.259)	(0.252)
Real estate					−0.200	−0.194
					(0.494)	(0.479)
Property value					0.185 *	0.179 *
					(0.098)	(0.095)
Constant	73.526 ***		360.585 ***		369.142 ***	
	(0.610)		(75.768)		(75.861)	
Birth cohort fixed effect	Yes	Yes	Yes	Yes	Yes	Yes
Community fixed effect	Yes	Yes	Yes	Yes	Yes	Yes
Observations	11,344	11,344	11,344	11,344	11,344	11,344

Note: The data in brackets are standard deviation, and *, **, and *** represent significance at the 10%, 5%, and 1% levels, respectively. Columns 1, 3, 5, and 7 show the estimation coefficients of the truncated regression model, and columns 2, 4, 6, and 8 show the average marginal effect.

**Table 4 ijerph-19-15530-t004:** Estimation results of the instrumental variable analysis.

Variable	Father		Mother	
	One-Stage	Two-Stage	One-Stage	Two-Stage
	(1)	(2)	(3)	(4)
Per capita number of Jinshi	0.171 ***		0.297 ***	
	(0.015)		(0.024)	
Children’s education		8.117 **		10.183 ***
		(3.949)		(3.948)
Control variable	Control	Control	Control	Control
Birth cohort fixed effect	Yes	Yes	Yes	Yes
Community fixed effect	Yes	Yes	Yes	Yes
Observations	7426	7426	8002	8002

Note: The data in brackets are standard deviation, and **, and *** represent significance at the 5%, and 1% levels, respectively. Columns 1–4 present the average marginal effect. The other control variables and constants are not reported in this table. Per capita number of Jinshi means the per capita number of Jinshi in each city of the Qing Dynasty (per 10,000).

**Table 5 ijerph-19-15530-t005:** Estimation results of the robustness tests.

**Panel A: Father**						
**Variable**	**Bootstrapping**	**Multiple Linear Regression**	**Cox**	**Truncation Regression**		
	**(1)**	**(2)**	**(3)**	**(4)**	**(5)**	**(6)**
Children’s education	0.943 ***	0.893 ***	−0.059 ***	0.698 ***		
	(0.170)	(0.151)	(0.012)	(0.162)		
Children’s years of education					0.265 ***	
					(0.038)	
Highest degree among parents’ children						0.875 ***
						(0.145)
Control variable	Control	Control	Control	Control	Control	Control
Birth cohort fixed effect	Yes	Yes	Yes	Yes	Yes	Yes
Community fixed effect	Yes	Yes	Yes	Yes	Yes	Yes
Observations	10,525	10,522	10,525	9236	10,525	10,188
**Panel B: Mother**						
**Variable**	**Bootstrapping**	**Multiple linear regression**	**Cox**	**Truncation regression**		
	**(7)**	**(8)**	**(9)**	**(10)**	**(11)**	**(12)**
Children’s education	0.774 ***	0.779 ***	−0.060 ***	0.675 ***		
	(0.139)	(0.137)	(0.012)	(0.172)		
Children’s year of education					0.196 ***	
					(0.036)	
Highest degree among parents’ children						0.852 ***
						(0.132)
Control variable	Control	Control	Control	Control	Control	Control
Birth cohort fixed effect	Yes	Yes	Yes	Yes	Yes	Yes
Community fixed effect	Yes	Yes	Yes	Yes	Yes	Yes
Observations	11,344	11,344	11,344	8360	10,500	10,977

Note: The data in brackets are standard deviation, and *** represent significance at the 1% levels, respectively. Columns 4, 5, 6, 10, 11, and 12 present the average marginal effect. The other control variables and constants are not reported in this table.

**Table 6 ijerph-19-15530-t006:** Results of the mechanism analysis.

Variable	Father			Mother		
	Emotional Support	Financial Support	Health	Emotional Support	Financial Support	Health
	(1)	(2)	(3)	(4)	(5)	(6)
Children’s education	0.602 ***	0.325 **	0.016	0.207 ***	0.241 ***	0.032 *
	(0.078)	(0.159)	(0.021)	(0.046)	(0.085)	(0.019)
Control variables	Control	Control	Control	Control	Control	Control
Birth cohort fixed effect	Yes	Yes	Yes	Yes	Yes	Yes
Community fixed effect	Yes	Yes	Yes	Yes	Yes	Yes
Observations	2243	2243	2243	3220	3220	2708

Note: The data in brackets are standard deviation, and *, **, and *** represent significance at the 10%, 5%, and 1% levels, respectively.

**Table 7 ijerph-19-15530-t007:** Results of the heterogeneity analysis.

	Father		Mother	
Variable	(1)	(2)	(3)	(4)
Children’s education	1.409 ***	0.654 ***	1.040 ***	0.641 ***
	(0.205)	(0.178)	(0.201)	(0.166)
Urban area × children’s education	−0.930 ***		−0.442	
	(0.290)		(0.283)	
Number of children × children’s education		0.761 ***		0.480 **
		(0.235)		(0.223)
Control variables	Control	Control	Control	Control
Birth cohort fixed effect	Yes	Yes	Yes	Yes
Community fixed effect	Yes	Yes	Yes	Yes
Observations	10,525	10,525	11,344	11,344

Note: The data in brackets are standard deviation, and **, and *** represent significance at the 5%, and 1% levels, respectively. Columns 1–4 present the average marginal effect. Other control variables and constants are not reported in this table. “×” indicates the cross-multiplication term of the two variables before and after.

**Table 8 ijerph-19-15530-t008:** Results for the analysis of parents-in-law’s longevity.

Variable	Father-in-Law		Mother-in-Law	
	(1)	(2)	(3)	(4)
Children’s education	0.467 ***	0.399 **	0.453 ***	0.460 ***
	(0.165)	(0.164)	(0.162)	(0.161)
Father-in-law’s education		1.151 ***		
		(0.228)		
Mother-in-law’s education				0.851 **
				(0.343)
Control variable	Control	Control	Control	Control
Birth cohort fixed effect	Yes	Yes	Yes	Yes
Community fixed effect	Yes	Yes	Yes	Yes
Observations	7503	7503	7399	7399

Note: The data in brackets are standard deviation, and **, and *** represent significance at the 5%, and 1% levels, respectively. Columns 1–4 present the average marginal effect. The other control variables and constants are not reported in this table.

## Data Availability

The datasets analyzed in this study are available from the China Health and Retirement Longitudinal Survey and Chinese Research Data Services Platform upon reasonable request.

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
