# Peer review of "Does Adult Children’s Education Increase Parents’ Longevity in China?"

_ijerph, 2022, doi:10.3390/ijerph192315530_

Round 1
Reviewer 1 Report
The topic of this manuscript is indeed innovative and at least to my knowledge there is not much research in the literature on this issue. Nevertheless I do have several concerns about this study which leads me to conclude that, in its present form, it would not be suitable for the research and academic community in general or for the international audience of IJERPH in particular.
I also have many concerns as to whether the issue addressed in this paper is appropriate for a journal of high academic quality such as ERR. I am saying this because the topic of the article is related to the field of education whereas the article submitted in the section Health Economics!
In my opinion, the paper seems to lack some critical steps in the explanation of key areas of the research, such as a thorough analysis of the research that would justify and develop the analysis with substantive explanations. There is no explanation of the criteria or the methodology used for such a quantitative analysis. I have no found the research questions or the research hypotheses of the study and how these affect the analysis. Hence, more clarity is needed on the actual design and implementation of the research.
After the description of the results the author(s) go(es) directly to the conclusion section, where I am left wondering what the real conclusion is. There is no discussion section that integrates the findings with the literature review, so as to provide the reader with a link between theory and practice. There are no limitations of the study nor suggestions are made for future research. Finally, I think that the article would certainly benefit from some proof reading / professional editing.
For the above reasons, in my opinion, the paper requires a substantial amount of work before it reaches a publishable standard.
Author Response
Please find the responses in the attachment.

Reviewer 2 Report
The strengths of the article are first of all the originality and importance of the topic, which is dealt with according to a very rigorous and precise methodology, obtaining very interesting results and research data. The weaknesses, although minor, mainly concern the first part (sections 1 and 2), in which it would have been appropriate to insert more theoretical references, in particular by further problematizing the concept of education and inserting references to a greater number of researches outside China. Nonetheless, the article still remains of high quality even in its current version.
Author Response
The strengths of the article are first of all the originality and importance of the topic, which is dealt with according to a very rigorous and precise methodology, obtaining very interesting results and research data.
Point 1:The weaknesses, although minor, mainly concern the first part (sections 1 and 2), in which it would have been appropriate to insert more theoretical references, in particular by further problematizing the concept of education and inserting references to a greater number of researches outside China.
Response 1: We appreciate your time and effort in reviewing our work and apologize for the confusion. The (i) concept of education, (ii) related theories, and (iii) references to studies conducted outside China have all now been added into the revised version of the manuscript. On point (i) above, we have added the following statement into footnote 1 on page 1: “Education in this study means formal education, a structured and systematic form of learning. Generally, this is education of a certain standard delivered to students by trained teachers in a designated place of education.” On point (ii), we have added a discussion on social network theory and social exchange theory into the manuscript as follows: “Based on social exchange theory [3], parents allocate resources to their children in their early years and adult children give back some of their resources to parents in later lifecycle stages” and “Social network theory [13] suggests that social networks impact health through four primary mechanisms: the provision of social support, social influence, social engagement and attachment, and access to resources and material goods.” On point (iii), we have added relevant references to studies performed outside China, including in the United States, South Africa, and Europe, as follows:
“Torssander [9] compared the different longevity of siblings using Swedish data. The results showed that sisters with children who have only received compulsory education are likely to die younger than sisters with children who have received higher education.”
“Friedman and Mare [10], using data from the United States, found that highly educated children increase the longevity of their parents, especially by reducing the risk of them dying from chronic respiratory and lung diseases.”
“De Neve and Harling [11] found that children’s increasing educational attainment is associated with a decline in the risk of parental death in South Africa.”
“Spitze and Logan [15] found that, at the same level of educational attainment, adult daughters contact their parents more frequently than do adult sons.”
Reviewer 3 Report
This is a very well written, readable article. It is not presented in the usual style Introduction, Background, Materials and Methods (Study design, participants, data collection and measures, and data analysis) then Results, Discussion, Strengths and Limitations, and Conclusions.
The way it is presented highlights the exemplary statistical analysis, the findings, then discusses those findings then moves on to the next analysis step. While it is an unusual, it made sense as I read it.
As a family gerontologist, the intergenerational transfers topic is quite important. The section which examines in-laws longevity was enlightening!
Overall this is a very readable, understandable paper.
Author Response
This is a very well written, readable article. It is not presented in the usual style Introduction, Background, Materials and Methods (Study design, participants, data collection and measures, and data analysis) then Results, Discussion, Strengths and Limitations, and Conclusions.
The way it is presented highlights the exemplary statistical analysis, the findings, then discusses those findings then moves on to the next analysis step. While it is an unusual, it made sense as I read it.
As a family gerontologist, the intergenerational transfers topic is quite important. The section which examines in-laws longevity was enlightening!
Overall this is a very readable, understandable paper.
Response: We appreciate your time and effort in reviewing our work. Thank you very much for your encouragement. Please note that further language changes have been made based on the comments of the other reviewers and additional professional editing work.
I revised the sections of introduction, data and methodology, conclusions and discussion.
We copied some changes of discussion as the follows.
Many studies explore the relationship between education and health [6,29]. At the same time, several economics studies have suggested that children’s education has a positive spillover effect on parents’ health [20,30]. This effect exists not only in China, but also in the United States [10], South Africa [11], and Europe [20]. Our research adds to the literature on the relationship between adult children’s human capital and parents’ health, and the results show that the intergenerational transfer between children and parents does not flow unidirectionally. Improving the degree of children’s education not only benefits children, but also benefits their parents.
The results of our study extend previous research on the relationship between children’s education and parents’ health or mortality risk [20,30]. One previous study examined the effect of children’s higher education on the mortality rate of parents with a low education level, concluding that the positive spillover effect of children’s education on younger elderly people was greater than that on older elderly people [20]. However, our research not only demonstrates the spillover effect of adult children’s education onto parents’ longevity but also concludes that the improvement in adult children’s education is beneficial to extending the longevity of parents-in-law. This finding encourages parents-in-law, who are an important part of the family, to connect with their sons- and daughters-in-law more often.
A limitation of our research is that the quality of the education of adult children is not considered because of the availability of data. The quality of adult children’s education is another important factor affecting parents’ longevity. The quality of the school to which adult children attended affects the quality of education. However, the name of the school attended is unavailable in our data. The second limitation is the lack of analysis of regional differences in living standards and life expectancy. In our data, parents’ birth dates range between 1900 and 1950, a period in which some regions experienced war. However. the effect of the experience of war on parents’ lifespan is not considered in our study. Meanwhile, De Neve and Fink [8] stated that the spillover effect of education on health differs by region. The third limitation is that we do not examine which of the three mechanisms is the most important; moreover, no other possible mechanisms are explored.
Nonetheless, the conclusions of this study suggest important policy implications. First, adult children’s education is an important factor affecting their parents’ longevity. A culture in which parents value their children’s education should thus be promoted. When a culture of higher education for children is cultivated, children are more likely to be willing to take on the responsibility of supporting their parents as they age. This can not only reduce endowment insurance pressure in China, but also improve the quality of parents’ lives as they become old. Second, family harmony should be promoted and a good family atmosphere should be created. Adult children’s education is beneficial to their spouse’s parents’ longevity as well as their own parents’ longevity. Parents are more likely to follow their children’s advice to increase their own healthy behaviors, instead of following the advice of other relatives. Therefore, family harmony should be advocated to increase the emotional connection between children and their parents. Finally, the elderly should be given training and support on the use of the mobile Internet. In the context of global aging, to reduce the gap between life expectancy in rural and urban areas and improve the quality of life among the left-behind elderly, access to and the usage of the Internet are important. Increasing network use training for the elderly would ensure that migrant children and the left-behind elderly can communicate via video when they are unable to meet frequently. As noted earlier, children increasing the emotional support they provide to their parents helps improve the latter’s physical and mental health. Education is thus the distribution of family resources and a means of addressing inequalities in family health. The increase in the education level of adult children can mitigate their parents’ lack of financial resources.
Further research should pay more attention to the extent to which the socioeconomic status of adult children affects their parents’ longevity and investigate how such an effect might differ across regions and populations as well as over time.
Round 2
Reviewer 1 Report
THANK YOU FOR REVISING THE PAPER. THE PAPER HAS BEEN MUCH IMPROVED